# Electromyographic Assessment of the Lower Leg Muscles during Concentric and Eccentric Phases of Standing Heel Raise

**DOI:** 10.3390/healthcare9040465

**Published:** 2021-04-14

**Authors:** Ukadike C. Ugbolue, Emma L. Yates, Kerensa Ferguson, Scott C. Wearing, Yaodong Gu, Wing-Kai Lam, Julien S. Baker, Frédéric Dutheil, Nicholas F. Sculthorpe, Tilak Dias

**Affiliations:** 1Faculty of Sports Science, Ningbo University, Ningbo 315211, China; guyaodong@nbu.edu.cn (Y.G.); jsbaker@hkbu.edu.hk (J.S.B.); 2Institute for Clinical Exercise & Health Science, School of Health and Life Sciences, University of the West of Scotland, South Lanarkshire G72 0LH, UK; emmayates21@hotmail.co.uk (E.L.Y.); kerensaferguson96@hotmail.co.uk (K.F.); nicholas.sculthorpe@uws.ac.uk (N.F.S.); 3School of Clinical Sciences, Queensland University of Technology (QUT), 60 Musk Avenue, Kelvin Grove 4059, Australia; s.wearing@qut.edu.au; 4Li Ning Sports Science Research Center, Li Ning (China) Sports Goods Co. Ltd., Beijing 101111, China; gilbertlam@li-ning.com.cn; 5Department of Kinesiology, Shenyang Sports University, Shenyang 110102, China; 6Centre for Health and Exercise Science Research, Department of Sport, Physical Education and Health, Hong Kong Baptist University, Hong Kong, China; 7CNRS, LaPSCo, Physiological and Psychosocial Stress, University Hospital of Clermont-Ferrand, CHU Clermont-Ferrand, Preventive and Occupational Medicine, WittyFit, Université Clermont Auvergne, 63000 Clermont-Ferrand, France; fred_dutheil@yahoo.fr; 8Faculty of Health, School of Exercise Science, Australian Catholic University, Melbourne 3000, Australia; 9Advanced Textiles Research Group, School of Art and Design, Nottingham Trent University, Bonington Building, Dryden Street, Nottingham NG1 4GG, UK; tilak.dias@ntu.ac.uk

**Keywords:** MVC, standing heel raise, heel rise, concentric muscle action, eccentric muscle action

## Abstract

Only a small number of muscle activation patterns from lower limbs have been reported and simultaneous muscle activation from several lower limb muscles have not yet been investigated. The purpose of this study was to examine any gender differences in surface electromyography (EMG) activity from six recorded lower limb muscles of the dominant limb at baseline (i.e., with the foot placed flat on the floor and in the neutral position), and during concentric and eccentric phases when performing a heel raise task. In total, 10 females and 10 males performed a standing heel raise task comprising of three continuous phases: baseline, unloading (concentric muscle action), and loading (eccentric muscle action) phases. Muscle activation from six muscles (gastrocnemius medialis, gastrocnemius lateralis, soleus, tibialis anterior, peroneus longus, and peroneus brevis) were measured using the Myon 320 EMG System. Root mean squared values of each muscle were calculated for each phase. Descriptive and inferential statistics were incorporated into the study. Statistically significant *p* values were set at 0.05. The results showed no significant differences between baseline, concentric, and eccentric phases with respect to each of the muscles investigated. Except for the gastrocnemius medialis at baseline and concentric phases, no significant differences were observed between genders or contractions. The data suggests that gender does not significantly influence the eccentric phase during the standing heel raise task.

## 1. Introduction

During human locomotion and everyday movements, muscular activity has been found to dampen soft tissue kinetic vibrations during each heel strike [1]. Lower limb muscles such as the gastrocnemius medialis (GM), gastrocnemius lateralis (GL), soleus (SO), tibialis anterior (TA), peroneus longus (PL), and peroneus brevis (PB) play an important role in absorbing kinetic energy when the foot makes contact with the ground [2]. These muscles also assist with foot stability and rigidity during weight bearing tasks including heel raises and push-offs [3]. The GM, GL, and SO have been observed to be the primary muscles in initiating the heel raise by contracting powerfully to lift the heel from the ground [4]. Previous research suggests that the PL and PB can provide gait stabilization functions by limiting inversion and controlling eversion when body mass is over the foot [5,6]. During each heel strike, the TA can prevent eversion of the foot by stabilizing the ankle [7]. Heel raises have many health benefits and are an important exercise for developing strength and stability of the lower extremity muscles [8]. Performing regular heel raise exercises can improve balance and muscle strength and thus reduce the incidence of falling [9]. The heel raise comprises an unloading phase where the lower leg muscles undergoes a concentric muscle action followed by a loading phase where the lower leg muscles undergo an eccentric muscle action. Although the muscle fiber type, architecture, function, and muscle co-activation vary across the GM, GL, SO, TA, PL, and PB lower limb muscles, understanding their electromyography (EMG) patterns during the heel raise task is of clinical importance and will benefit practitioners in terms of designing exercise regimes that could potentially translate into greater training adaptations and injury management [10]. A study by Riemann and associates revealed that when the foot was positioned in neutral (i.e., pointing forward and not internally or externally rotated) during the heel raise exercise, the GM and GL both showed no significant differences during the concentric and eccentric activation phases [10]. In a walking study on a flat surface at a self-selected speed, Zdolšek et al. showed that during the toe-off phase of gait, a small EMG signal amplitude for the TA was produced in comparison to the GM, GL, SO, PL, and PB lower limb muscles [11].

Several factors such as age, foot position, and leg dominance may influence muscle activity within the lower extremities [3,12,13]. Compared to the non-dominant limb, higher joint torque generation and muscle strength have been associated with the dominant limb during a heel-raise task [14]. However, studies investigating the influence of leg preference in non-athletes have indicated that there were no asymmetries when comparing muscle power, flexibility, and balance within both limbs [12,15]. Therefore, due to symmetrical functional characteristics being presented for both legs, this study aimed to analyze the muscular activity of the dominant leg.

Research by von Tscharner and Goepfert highlighted that there may be specific gender differences in muscular activity within the GM and TA during distinct running phases [16]. Furthermore, during every day physical movements such as a heel raise, relative EMG values within the triceps surae muscles (GM, GL, and SO) have been found to be significantly higher in young females compared to young males [13]. However, previous research has indicated that there were no significant gender differences in muscle activation in the calf muscles during other tasks such as double-leg hopping [17]. The difference in muscle activation between males and females has been linked to force production, strength, and neuromuscular control [18,19]. It has been suggested that females may compensate for reduced force production and strength by increasing the EMG activation of the calves and quadriceps [20]. In addition, the majority of the literature has only examined the activation of a small number of muscles within the lower leg between genders [3,21], and little attention has been paid to investigating all muscles in the lower limb muscle compartments (i.e., TA, GM, GL, SO, PL, and PB). Therefore, the purpose of this study was to examine any gender differences in the surface EMG activity of all lower leg muscles of the dominant limb at baseline (i.e., with the foot placed flat on the floor and in the neutral position), and during the concentric and eccentric phases of the heel raise. We hypothesize that the lower leg muscles would exhibit significant differences in the muscle activation phases of the heel raise and that these differences would vary across genders.

## 2. Methods

### 2.1. Participants

The work reported here was granted ethical approval by the School of Health and Life Sciences ethics committee (approval number 2017-0967-844) at the University of the West of Scotland. The principles outlined in the Helsinki Declaration were followed. The eligibility criteria were as follows: To fulfill the inclusion criteria all participants had to be healthy and aged over 16 years old. They also had to be physically active. All participants under the age of 18 years old were granted parental consent before embarking onto the study. Participants were excluded from the study if they had any lower limb asymmetry (i.e., morphological and/or functional), history of lower limb musculoskeletal injuries or surgery, physical disabilities, foot arch and toe deformities, pathology, and/or pregnant. The physical activity level of all participants recruited were low to moderate. However, participants with higher physical activity levels were also considered eligible to participate in the study. The American College of Sports Medicine (a complete guide to fitness and health) has highlighted that a target of 150 min/wk in terms of duration and frequency of exercise was considered as a moderate-intensity level for maintaining good health [22]. Following ethical approval, 20 healthy participants volunteered for this laboratory-based experiment. All participants signed informed consent forms and were made fully aware of experimental procedures prior to data collection. The volunteers consisted of ten males (age 26.3 ± 11.7 years, height 180.2 ± 4.5 cm, body mass 78.7 ± 10.3 kg, right ankle width 77.0 ± 6.7 mm; mean ± SD), and ten females (age 22.3 ± 11.5 years, height 164.3 ± 6.0 cm, body mass 57.5 ± 10.1 kg, right ankle width 71.0 ± 4.6 mm; mean ± SD). All participants were right leg dominant and this was confirmed by asking the participants to kick a ball at a target placed 4 m away as described previously [23].

### 2.2. Equipment

The AMTI force plates (AMTI, Watertown, MI, USA), Vicon Nexus Bonita motion system (Vicon Nexus Bonita, Oxford Metrics Ltd., Yarnton, Oxfordshire, UK) and the wireless Myon 320 EMG device (Myon 320, Myon AG, Schwarzenberg, Switzerland) were synchronized to record the kinetic, kinematic and EMG activity throughout the heel raise task. Two force plates, BP400600AMTI Optima Human Performance System (AMTI, Watertown, MI, USA) sampling at 1 kHz were embedded in concrete and used in the analysis. The kinetic outputs ensured that the ground reaction forces on the force plates between both limbs were uniformly distributed and equal. An eight-camera motion analysis (Vicon Nexus Bonita, Oxford Metrics Ltd., Yarnton, Oxfordshire, UK) system with a sampling rate of 250 Hz was positioned around the participant’s heel. A marker set consisting of thirteen retroreflective markers (3 mm) were placed on the dominant heel to track each phase of the heel raise (baseline, concentric and eccentric) during EMG recordings. The position and orientation of the cameras was marked to standardize the view of the heel. Muscle activity was recorded by eight surface EMG Transmitters (Myon 320, Schwarzenberg, Switzerland) and surface electrodes (AMBU Ltd, Alconbury Weald, Cambridgeshire, UK). 

### 2.3. Retroreflective Marker Setup

A custom designed template was used to position the retroreflective markers on the heel. The flexible flat plastic designed template consisted of 13 provisions arranged in three rows (Figure 1I). The first row consisted of three provisions for three retroreflective markers. Each provision had a horizontal spacing of 1.5 cm between each provision. The second row consisted of three provisions for three retroreflective markers. In addition, each provision on the second row had a horizontal spacing of 1.5 cm between each provision. The third row consisted of seven provisions for seven retroreflective markers. These seven provisions were positioned 1 cm from the bottom edge of the template such that this region of the heel represented the heel pad. The vertical spacing distance on the template between the rows was 2 cm.

Participants were asked to stand barefoot with their weight distributed equally on both feet while the template was placed on the heel. The location of the template was marked with a pen and tape and was transferred to the labelled areas on the skin (Figure 1-II). Retroreflective markers were placed at three different levels: top layer (TOP), middle layer (MID), and lower layer (LOW). Seven markers were placed along the lower circumference of the fat pad, three markers were positioned on the middle section and three markers were attached to the upper segment of the heel (Figure 1).

### 2.4. EMG Procedure

Electrodes were placed over the belly of the muscles under investigation. To reduce the impedance interface between the skin and the electrodes, the tested areas were shaved and cleaned using abrasive pads for skin preparation (AMBU ^®^ 2121M Skin Prep Pad, Ambu Ltd., Cambridgeshire, UK) and alcohol wipes (Medi-Swab Seton Health Care Group Plc, Banbury, UK). According to SENIAM guidelines [24] for the lower leg, electrodes with an inter-electrode separation of 2 cm were placed on the longitudinal axis of the muscle belly of the six muscles on the dominant leg: gastrocnemius medialis (GM), gastrocnemius lateralis (GL), soleus (SO), tibialis anterior (TA), peroneus longus (PL), and peroneus brevis (PB) (Figure 2). 

To standardize this procedure, the same chartered physiotherapist reviewed the placement of all surface EMG electrodes for all participants. Adhesive tape was used to secure the EMG electrodes and monitors to the leg surface. The participants performed three maximum voluntary contractions (MVC) for two seconds [25] with a 2 min rest period between tests. In brief, combined movements of ankle dorsiflexion and foot inversion while sitting were used to measure the MVC for the TA. To measure the MVC for the PL and PB participants, we performed foot eversion with relaxed toes whilst sitting with the knee extended. The MVC for the GM and GL was measured when standing with the ankle in plantarflexion and the knee extended. Additionally, the MVC for the SO was measured in a standing position with the ankle in plantarflexion and the knee flexed. Using the proEMG software (Myon 320, Schwarzenberg, Switzerland), EMG data were processed using a high-pass filter set at 20 Hz and a low-pass filter set at 450 Hz. The sampling frequency was 1 kHz and pre-amplified. The settings within the pre-amplifier included an input impedance of 2 MΩ and a gain of 1000. To analyze the EMG recordings of the heel raise, data was presented as root mean squared (RMS) with a moving window of 200 ms [3]. EMG signals were further processed in Microsoft Excel 2017 version 16.10 (Microsoft Corporation, Redmond, Washington, DC, USA).

### 2.5. Experimental Design

Participants were required to visit the Biomechanics Laboratory at the University of the West of Scotland on one occasion and data was collected between the hours of 9 and 11 in the morning to minimize any diurnal variation influences on data collection. After the anthropometric measurements (height, mass, and ankle width) were taken, each participant was given a 10 min familiarization period to practice a slow and controlled heel raise. Body mass, height, and ankle width were recorded to the nearest 0.1 cm using clinical scales (Seca 803, Seca GmbH, Hamburg, Germany), a stadiometer (Seca 213, Seca GmbH, Hamburg, Germany) and a 150 mm measuring capacity slide-knot caliper (Mediworld, London, UK), respectively.

Participants were asked to stand in front of the cameras with their feet in a neutral position, a shoulder-width apart with hands placed on the hips. Participants were instructed to slowly raise both heels as high as possible and lower both heels evenly on to the ground. The heel raise comprised of three continuous phases namely the baseline phase, concentric phase, and eccentric phase [26,27,28]. The transitions between the concentric and eccentric phases were the same. Each phase lasted for two seconds and was verbally counted and recorded by the same practitioner. Specifically, at the baseline phase and upon attaining the end plantarflexion range positions at the concentric and eccentric phases participants held the phase positions for two seconds. Three trials were recorded, and a 1 min rest period was provided between each trial. This minimized the likelihood of any potential fatigue effect building up between trials [26,27,28]. With respect to each participant the EMG measurement between trials were indeed very close and this was also in agreement with the kinematic and kinetic outputs. The heel phase positions were inspected both numerically (i.e., based on force outputs) and visually. These inspections were reliably checked and validated using the motion capture system (Vicon Nexus Bonita, Oxford Metrics Ltd., United Kingdom). 

### 2.6. Data Analysis

Three dimensional videos and images from the motion capture system allowed the data analysts to identify the baseline, concentric, and eccentric phases. These heel positions were held for two seconds at each phase and then confirmed by two researchers prior to recording the maximum muscle activation during the two second window with respect to each phase. The data was averaged from the three trials. Descriptive statistics including means and standard deviations (SD) were calculated for all six tested muscles during each phase of the heel raise (baseline, concentric and eccentric). The following equation was used to calculate muscle activity: $$\frac{\mathrm{Dynamic}\;\mathrm{EMG}\;\mathrm{data}\;\left(\mathrm{heel}\;\mathrm{raises}\right)}{\mathrm{MVC}}\times 100$$

Muscle activity was expressed as a percentage of the MVC. A test for normality using the Shapiro-Wilk test was conducted. The outcome of the test revealed that the data was not normally distributed and therefore non-parametric tests were required to statistically evaluate the data set. The Mann Whitney U tests were used to evaluate gender differences for muscle activation during each phase of the heel raise. All muscle activation data were post processed during each phase of the heel raise using Microsoft Excel 2017 version 16.10 (Microsoft Corporation, Redmond, Washington). 

Chi-Square (*X^2^*), degrees of freedom (*df*), and asymptotic significance level were obtained. Significance was set at 0.05. Pairwise differences expressed as Mann–Whitney tests were performed for the condition grouping variable (Phase effect) with respect to all six tested muscles whilst performing the Kruskal–Wallis H test. Significant values for the pairwise difference was adjusted by a Bonferroni correction to avoid inflation of Type 1 error. The adjusted significance level was set to *p* = 0.017.

## 3. Results

The Shapiro–Wilk test for normality revealed the muscle activity from all six muscles (TA, GM, GL, SO, PL, and PB) were not normally distributed (*p* < 0.007). Although no significant differences between the male and female participants were observed for the BMI (*p* = 0.060) output, there were significant differences with respect to gender for body mass (*p* < 0.001).

Overall, the grouping variable gender showed only a significant difference for the GM muscle (*p* < 0.001). When comparing gender, the TA muscle activity presented no significant differences between males and females during the baseline phase (*p* = 0.597). The muscle activity of the GM produced significant differences between genders during the baseline phase (GM: *p* = 0.023). However, the GL showed no significant differences between genders during the baseline phase (GL: *p* = 0.345). There were no significant differences in muscle activity in the SO between males and females during the baseline phase (*p* = 0.705). The PL and PB muscle activity showed no significant differences between genders during the baseline phase (PL: *p* = 0.762; PB: *p* = 0.821) (Figure 3).

The TA muscle activity presented no significant differences between males and females during the concentric phase (*p* = 0.496). The muscle activity of the GM showed significant differences between genders during the concentric phase (GM: *p* = 0.041). The GL showed no significant differences between genders during the concentric phase (GL: *p* = 0.212). There were no significant differences in muscle activity in the SO between males and females during the concentric phase (*p* = 0.496). The PL and PB muscle activity showed no significant differences between genders during the concentric phase (PL: *p* = 0.597; PB: *p* = 0.705) (Figure 4). 

The TA muscle activity presented no significant differences between males and females during the eccentric phase (*p* = 0.545). The muscle activity of the GM and GL showed no significant differences between genders during the eccentric phase (GM: *p* = 0.054; GL: *p* = 0.545). There were no significant differences in muscle activity in the SO between males and females during the eccentric phase (*p* = 0.650). The PL and PB muscle activity showed no significant differences between genders during the eccentric phase (PL: *p* = 0.880; PB: *p* = 0.880) (Figure 5).

The nonparametric Kruskall–Wallis test revealed no significant differences for condition (phases). Specifically, the hypothesis test summary showed the null hypothesis was retained based on the following outputs: TA (*p* = 0.956, *X^2^* = 0.091, *df* = 2), GM (*p* = 0.914, *X^2^* = 0.180, *df* = 2), GL (*p* = 0.915, *X^2^* = 0.177, *df* = 2), SO (*p* = 0.908, *X^2^* = 0.193, *df* = 2), PL (*p* = 0.991, *X^2^* = 0.017, *df* = 2), and PB (*p* = 0.930, *X^2^* = 0.145, *df* = 2).

The pairwise test results showed no significant differences between baseline and concentric phases with respect to the TA (*p* = 0.787), GM (*p* = 0.655), GL (*p* = 0.685), SO (*p* = 0.914), PL (*p* = 0.946), and PB (*p* = 0.695) muscles. 

The pairwise test results revealed no significant differences between baseline and eccentric phases with respect to the TA (*p* = 0.935), GM (*p* = 0.776), GL (*p* = 0.957), SO (*p* = 0.695), PL (*p* = 0.935), and PB (*p* = 0.946) muscles.

The pairwise test results showed no significant differences between the concentric and eccentric phases with respect to the TA (*p* = 0.808), GM (*p* = 0.914), GL (*p* = 0.756), SO (*p* = 0.725), PL (*p* = 0.903), and PB (*p* = 0.808) muscles.

The %MVC data for all the 20 participants is displayed in Figure 6.

## 4. Discussion

The present study investigated the EMG activation of the lower limb muscles (TA, GM, GL, SO, PL, and PB) between males and females during baseline, concentric, and eccentric phases of a heel raise. Our findings showed that there were no significant changes in muscle activation of all six muscles during each phase of the heel raise. Although our results did not agree with the hypothesis presented, with respect to gender, the GM produced significant differences at baseline and concentric conditions. Compared to all six tested muscles, the TA portrayed a higher mean %MVC within both genders at baseline, concentric, and eccentric phases. From our results, three females presented with a TA %MVC of 81%, 82%, and 91%, respectively whereas two males presented with a TA %MVC of 126% and 130%, respectively. These high percentage maximum voluntary contractions caused the TA outputs to be on average 54% and 52% for both males and females, respectively. Without these outputs the TA %MVC would have been 35% and 38% for males and females, respectively. To date, there is no published data investigating the gender differences for healthy participants in TA muscle activation during a heel raise. Research by André et al. (2018) indicated that EMG activity for the TA within elderly participants remained small (2.0 ± 0.9 %MVC) during an entire heel raise cycle. As a result, the higher muscle activation within the TA for both genders may be due to the heel raise being performed at a slow and controlled pace in our testing protocol. This could have resulted in the TA constantly having to stabilize the ankle because of the slower movement pattern [29]. 

The muscle activation of the plantar flexors (GL, GM, and SO) were quite low during the baseline, concentric, and eccentric phases of the task. This outcome could be attributed to several physiological related reasons. Most notably the surface area of the muscle belly where the surface EMG was placed may have produced low muscle activation patterns. This may have resulted in small and weakly fired motor units which unfortunately did not progressively lead to the recruitment of larger units during the transitions between the baseline, contraction and eccentric phases measured. The results showed that the mean %MVC for female GL muscle activation was higher at each phase of the heel raise, however, no significant differences between genders was highlighted. On the other hand, mean %MVC for male GM muscle activation was higher than females for each phase of the heel raise. Furthermore, findings from the present study suggest that there were no significant differences in GL muscle activation between males and females during the baseline, concentric, and eccentric phases of the heel raise. Similar results were portrayed by Padua et al. (2005), who found that females had a greater GM and GL muscle activation during a two-footed hopping task, even though no significant interactions were demonstrated. von Tscharner and Goepfert (2003) found distinct gender differences in surface EMG signals for the GM within runners and further suggested that the changes in muscle activation were significant enough to distinguish participants as either female or male. It is worth considering that von Tscharner and Goepfert (2003) analyzed GM activation during running which requires different motor recruitment patterns compared to a heel raise. Therefore, gender differences in EMG activation within the gastrocnemius muscles may be associated with increasing intensity or frequency of muscle activation performed in movements such as running [30].

In this study, mean %MVC SO muscle activation was slightly higher in females than males during each phase of the heel raise. Likewise, research by Padua et al. (2005) indicated that SO muscle activity was 38% greater in females compared to males during two-footed hopping. Despite this, there were no significant differences in SO muscle activation between males and females within this present study. It is proposed that the small difference in SO muscle activation may be due to feedforward neuromuscular control by which females compensate for reduced force production, strength and lower extremity stiffness by increasing the activation of the SO muscles for dynamic joint stability [20]. 

This study also reported no significant changes in PL and PB muscle activation during each phase of the heel raise. Additionally, mean %MVC muscle activation in both the PL and PB was higher in males compared to females at each phase of the heel raise. The findings from this study agree with previous research, which investigated the PL and PB in walking conditions; however, no research analyzing the PB during a heel raise was found. Mueller and associates showed that PL EMG activity was slightly higher in males during normal walking [31]. Controversially, Mueller et al. (2018) also indicated that females displayed significantly greater PL activity than men when walking with provoked stumbling. These results are similar to Baur et al. who demonstrated that female competitive runners had a higher PL muscle activation just before the heel made contact with the ground when running on a treadmill [32]. EMG muscle activation has been shown to heighten in the peroneal muscles during single-footed support stance or unstable movement due to their stabilizing function during gait [33,34]. Therefore, it could be possible that peroneal muscle activation during this study may have been influenced by the two-footed heel raise stance which provided a stable platform for the ankle muscles [3]. The previous literature suggests that the PL and PB may have a gender specific role in controlling ankle joint stability during activities which involve an entire gait cycle such as walking or running [32]. More recently, research by Ugbolue et al., in their novel methodological approach to evaluate heel pad stiffness and soft tissue deformation among males and females, showed that on average males produced stiffer heel pads than females during the loading and unloading phases [27,28]. Nonetheless, the influence of gender on the PB is still unclear and further research is required to understand this relationship. Thus, the findings here are incremental and together with our earlier study on heel pad stiffness and soft tissue deformation contribute towards scientific advances in this area of research.

Inter and intra diurnal variation effects [35,36] were minimized for the 20 participants, as time of data collection was standardized for all subjects. Muscle activity in the PL from one participant for all phases of the heel raise was outlying, therefore this was removed from the analysis to prevent the data set from being affected. Unusual and abnormally high %MVCs that did not correspond to the %MVC pattern across the 20 participants were considered outliers. Although one outlier was removed from the data analysis three females and two male outliers at the baseline phase and two males and two females as outliers for the concentric and eccentric phases respectively were present in the data set. This variability is highlighted in Figure 5 and presented as a representative scatter plot showing the %MVC data set for all participants. A potential limiting factor for this present study was that surface EMG was used to analyze all six muscles in the lower limb. Incorporating fine wire/needle EMG into this current study may have resulted in a more precise insertion of the electrodes on smaller muscle bellies such as the TA, PL, and PB. Additionally, only the dominant limb was analyzed within this study. It cannot be ruled out that leg dominance along with foot posture may have influenced muscle activation during the heel raise protocol. Moreover, whilst the bar charts reported in the figures are reflective of the presented results, it is evident that the standard deviations were high. This outcome may have occurred for a variety of reasons. The participants may have exhibited some natural differences due to muscle physiology (e.g., number of motor units present in a specific muscle region detected by the EMG electrode capture volume). Other likely sources for variability in EMG recordings between participants, mostly relate to methodological issues such as problems in the normalization process, signal processing, skin impedance, crosstalk, etc. Although precautionary measures were observed during the participant preparation, experimental setup and data collection sessions, it is noteworthy that an appropriate normalization usually reduces inter-subject variability [37]. Further, during experimental planning the authors did not incorporate consideration for measurements acquired in accordance with the menstrual cycle of the female participants. It has been demonstrated that lower limb muscles in females are differentially recruited across the menstrual cycle [38,39,40]. This may be another limitation of this study. A further important consideration is the fact that, although no significant differences were observed for the BMI outputs (*p* > 0.05), statistical differences were exhibited between genders with respect to body mass (*p* < 0.001). Furthermore, despite the low to moderate physical activity levels demonstrated by the participants, it is probable that the results may have been influenced by the differences in the physical activity levels of the participants.

The absolute variability and relatively high dispersion from the mean in the EMG data from the outliers may have influenced the large standard deviations reported. The higher dispersion or variability in the muscle activity was only presented by a small number of participants with respect to each of the six muscles. All acquired muscle activity data sets were objectively justified and correctly normalized with respect to the MVC outputs from each of the participants and their respective muscles. While a limitation of this study might be the moderate sample population of 10 men and 10 women, more recently a study by Alizadeh et al. also used a moderate sample population of 10 men and 10 women [41]. Our moderate sample size of 20 participants is justified and was based on a sample size calculation. A preliminary feasibility study was conducted in our biomechanics laboratory and the outputs were used to generate the initial power analysis. As this study is a preliminary feasibility study, we chose a minimal sample size of 10 participants from each group. Thus, with a sample size of 10 males and 10 females a power of 80% was achieved. Moreover, frequency domain of the EMG was not analyzed and investigated in this study. Lastly, although the motor task was related to the push off phase of walking and running, a future study on gait analyses is planned to better understand how to translate and relate our results to a functional activity of daily living task such as walking. This would provide more clarity into recognizing the muscle activation outputs particularly at heel strike and toe-off gait events and observing how they vary from a static standing toe raise task and a static standing heel raise task, respectively. 

To summarize, based on the findings of this study, gender does not seem to significantly influence the concentric or eccentric phase of a heel raise performed at a slow and controlled speed. Therefore, it is envisaged that from a non-invasive methodological approach, EMG outputs in combination with kinetic and kinematic outcome measures could be used as a screening tool that provides valuable information in the assessment of muscles and soft tissues as they undergo rehabilitation and recovery following injury. 

## Figures and Tables

**Figure 1 healthcare-09-00465-f001:**
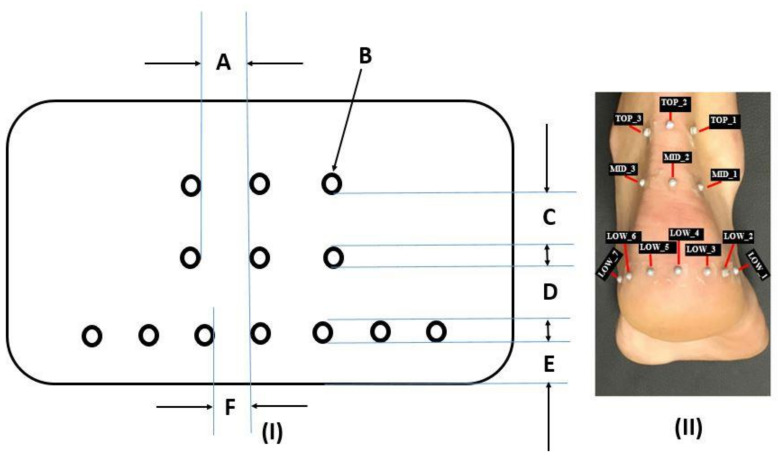
(**I**)**:** Custom designed template used to position the retroreflective markers on the heel. The internal dimensions were as follows: A represents the horizontal spacing distance for the TOP and MID retroreflective marker provisions (2 in number), B represents the 2 mm diameter perforated holes (13 in number), C represents the 2 cm vertical spacing distance between the TOP and MID retroreflective marker provisions, D represents the 2 cm vertical spacing distance between the MID and LOW retroreflective marker provisions, E represents the horizontal spacing distance for the LOW (bottom) markers (6 in number). (**II**): Pictorial illustration showing the retroreflective marker positions on the heel (Note: the size of the retroreflective markers are 3 mm in diameter).

**Figure 2 healthcare-09-00465-f002:**
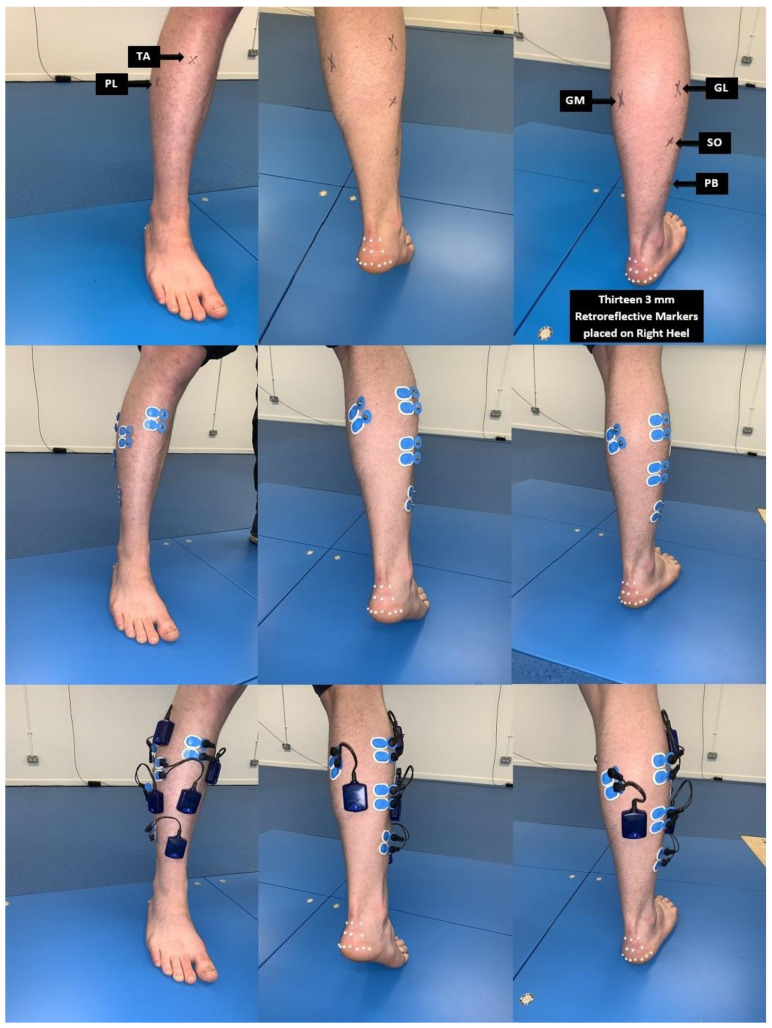
**Row 1:** Presented on the right leg are marked positions of the lower limb muscles. **Row 2:** The marked positions are covered with self-adhesive AMBU surface electrodes positioned on the GM, GL, SO, TA, PL and PB. **Row 3:** The self-adhesive AMBU surface electrodes were connected to the Myon 320 transmitters (transmitters were secured to the calf muscles with double-sided tape). The columns represent the baseline, unloading (concentric) and loading (eccentric) phases of the heel raise task, respectively. Retroreflective markers on the heel were used for tracking the heel phase positions.

**Figure 3 healthcare-09-00465-f003:**
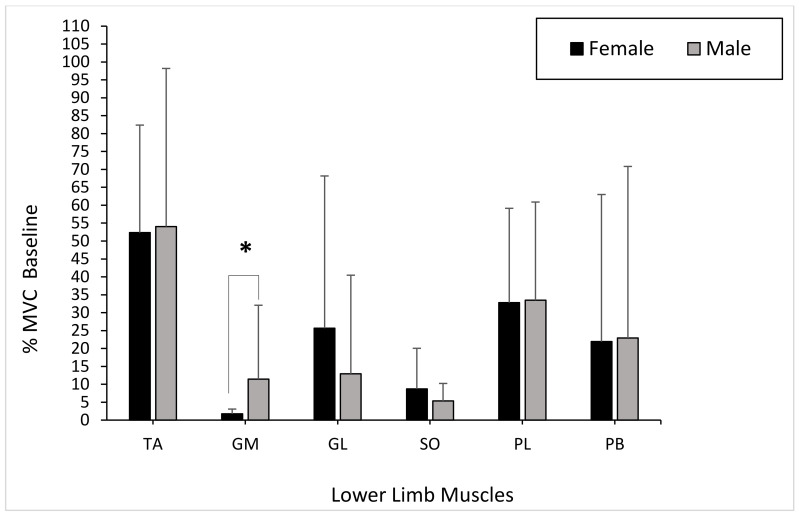
Mean gender differences in muscle activation (gastrocnemius medialis (GM), gastrocnemius lateralis (GL), soleus (SO), tibialis anterior (TA), peroneus longus (PL), and peroneus brevis (PB)) within the dominant limb during the baseline phase with error bars (± standard deviation; * statistical significance).

**Figure 4 healthcare-09-00465-f004:**
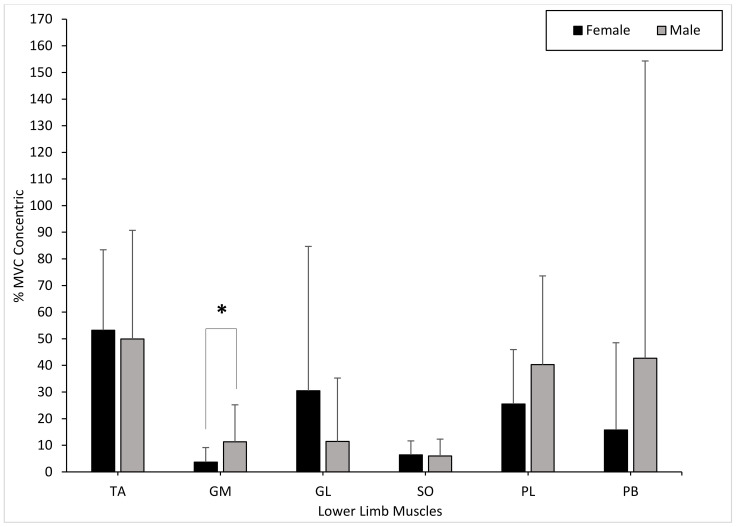
Mean gender differences in muscle activation (TA, GM, GL, SO, PL, and PB) within the dominant limb during the concentric phase with error bars (± standard deviation; * statistical significance).

**Figure 5 healthcare-09-00465-f005:**
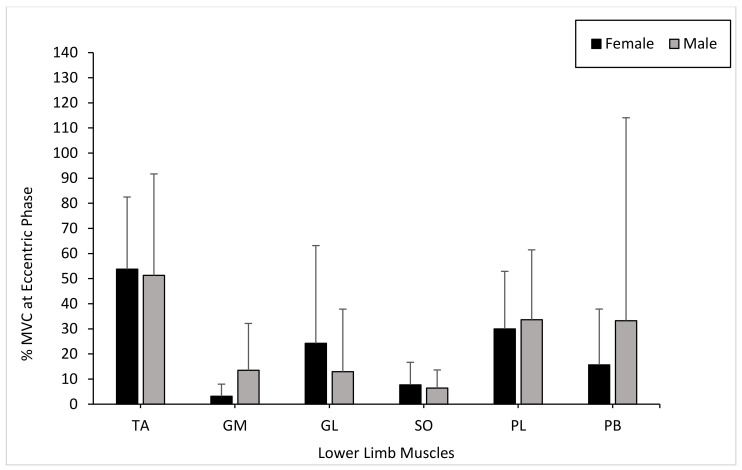
Mean gender differences in muscle activation (TA, GM, GL, SO, PL, and PB) within the dominant limb during the eccentric phase with error bars (± standard deviation).

**Figure 6 healthcare-09-00465-f006:**
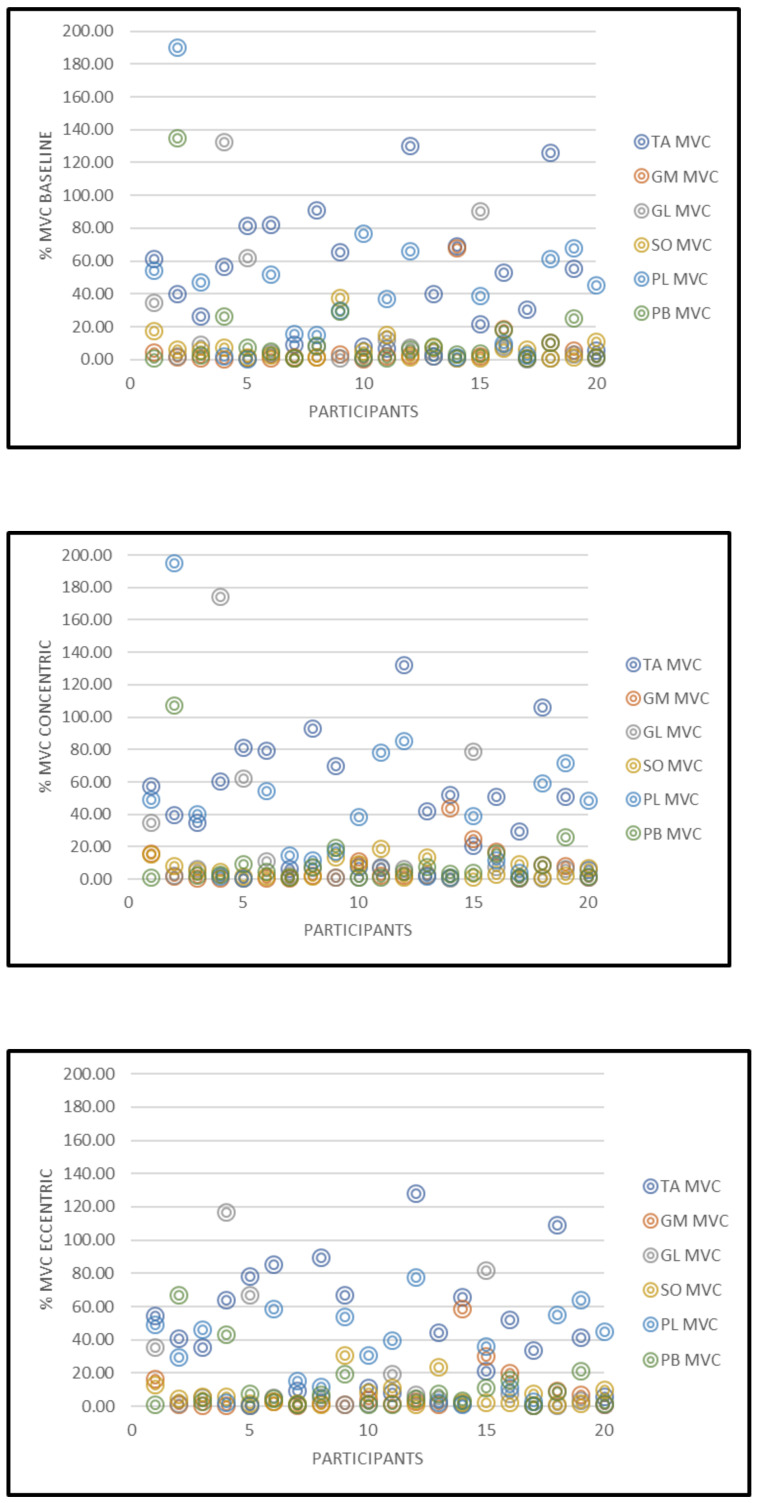
Representative data set showing the percentage maximum voluntary contractions scatter plots at baseline, concentric, and eccentric phases for all 20 participants.

## Data Availability

The data presented in this study are available on request from the corresponding author. The data are not publicly available due to privacy restrictions.

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
