# Peer review of "Electromyographic Assessment of the Lower Leg Muscles during Concentric and Eccentric Phases of Standing Heel Raise"

_healthcare, 2021, doi:10.3390/healthcare9040465_

Round 1

Reviewer 1 Report

My comments to the authors are:

The main problem with this study is that the sample size is very small, only 10 subjects per group. Authors should add sample size calculation.

Furthermore, the authors should show if there were differences between the two groups. For example, the weight and the BMI could be significantly different and therefore affect the results.

Author Response

Dear Reviewer,

Thank you very much for your comments. We have now addressed them and incorporated the changes to our manuscript. Below are the responses to your comments

The main problem with this study is that the sample size is very small, only 10 subjects per group. Authors should add sample size calculation.

Response: Thank you for your comment. We used 20 participants altogether and we agree that the sample size is small. A preliminary feasibility study was conducted in our biomechanics laboratory and the outputs were used to generate the initial power analysis. As this study is a preliminary feasibility study we chose a minimal sample size of 10 participants from each group. Thus, with a sample size of 10 males and 10 females a power of 80% was achieved. This statement has been added to our discussion.

Furthermore, the authors should show if there were differences between the two groups. For example, the weight and the BMI could be significantly different and therefore affect the results.

Response: This is an important point. A statement on the body mass and BMI outputs are included in the results and discussion sections to reflect your concern. Although no significant differences between the male and female participants were observed for the BMI (p = 0.060) output, there were significant differences with respect to body mass (p < 0.001) between the males and females.

Kind regards,

U. Chris Ugbolue, PhD

Reviewer 2 Report

Dear authors,

You have written an interesting paper, however, some parts need to be addressed for greater clarity and reproducibility.  

Introduction

Lines 57 to 63 need references to bask up your claims

Methods:

  • Where the principles of the Helsinki declaration followed? Report
  • Report the ethical approval document number
  • What were the inclusion and exclusion criteria? Report
  • What was the physical activity level of these participants (as this can affect your results - were they athletes, active population,...-)? Report
  • You have a big SD in the age of both genders. Elaborate?
  • Connected to the big SD in age - have you acquired the written consents of parents or guardians for those under 18? Report

Lines 120-122 - what was the distance between those 13 markers - what was the ''anchoring point'' and the distance between them in each line and between the lines? Report

Lines 164-167 - Report how and whit which instruments were the height, body mass, and ankle width measured.

Lines 172-173 - Back up your decision with references of 2s for each phase and not 3 or more. Would this provoke a higher activation? Additionally, the word approximately was used. So did they perform each stage for 2s or not? This can be understood as some did it for 1.5s and others maybe 1.8s. Elaborate

Lines 175-176 - Reference needed to back up your decision of a 1-minute rest.

Line 188 - What did you do if there were 2 results close together and the 3rd had a significant difference? Did you still take the average of 3 reps or did you perform the 4th rep as this could impact your data? Elaborate

Results

Figure 2 - Significance in GM not marked - correct

Overall your results show high SD which brings me to your sample selection and its size. Especially in PB con and ecc measurements. Elaborate

Limitations of the study should be extended with the difference in the physical activity levels of participants.

What is the practical application of your findings for the field? Add

Kind regards

Author Response

Dear Reviewer,

Thank you very much for your comments. All comments have now been addressed and the changes have been incorporated into our manuscript. Below are the responses to your comments.

Introduction

Lines 57 to 63 need references to bask up your claims

Response: Thank you for your comment. Two references have now been added to the end of lines 57-63.

Methods:

  • Where the principles of the Helsinki declaration followed? Report

Response: The principles of the Helsinki declaration were followed and all ethical principles and guidelines were applied to our research.

  • Report the ethical approval document number.

Response: The Ethical approval number is 2017-0967-844. This has now been added to the section 2.1 of the Methods section.

  • What were the inclusion and exclusion criteria? Report

The eligibility criteria were as follows. To fulfil the inclusion criteria all participants had to be healthy and aged over 16 years old. All participants under the age of 18 years old were granted parental consent before embarking onto the study. Exclusion Criteria – Participants with any lower limb asymmetry, lower limb musculoskeletal injuries, physical disabilities, pathology and/or pregnant were excluded from the study. This statement has been added to section 2.1.

  • What was the physical activity level of these participants (as this can affect your results - were they athletes, active population,...-)? Report

Response: The physical activity level of all participants were low to moderate. This is also added to section 2.1.

  • You have a big SD in the age of both genders. Elaborate?

Response: You are correct we had a wide age range for both genders. For the males the age range was 16 – 49 years while the age range for the females was 16 – 54 years. We hope this explains why the SD is large.

  • Connected to the big SD in age - have you acquired the written consents of parents or guardians for those under 18? Report

Response: As indicated earlier parental consent was acquired for participants under the age of 18 years.

Lines 120-122 - what was the distance between those 13 markers - what was the ''anchoring point'' and the distance between them in each line and between the lines? Report

Response: A detailed description of the dimensions have been added to the methods section. The following paragraphs have been incorporated into the word document together with a new figure. See below:

A custom designed template was used to position the retroreflective markers on the heel. The flexible flat plastic custom designed template consisted of 13 provisions arranged in three rows (Figure 1-I). The first row consisted of three provisions for three retroreflective markers. Each provision had a horizontal spacing of 1.5 cm between each provision. The second row consisted of three provisions for three retroreflective markers. In addition, each provision on the second row had a horizontal spacing of 1.5 cm between each provision. The third row consisted of seven provisions for seven retroreflective markers. These seven provisions were positioned 1 cm from the bottom edge of the template such that this region of the heel represented the heel pad. The vertical spacing distance on the template between the rows was 2 cm.

Participants were asked to stand barefoot with their weight distributed equally on both feet while the template was placed on the heel. The location of the template was marked with a pen and tape was transferred to the labelled areas on the skin (Figure 1-II). Retroreflective markers were placed at three different levels; top layer (TOP), middle layer (MID) and lower layer (LOW). Seven markers were placed along the lower circumference of the fat pad, three markers were positioned on the middle section and three markers were attached to the upper segment of the heel (Figure 1).

Lines 164-167 - Report how and whit which instruments were the height, body mass, and ankle width measured.

Response: Body mass, height and ankle width were recorded to the nearest 0.1cm using clinical scales (Seca 803, Seca GmbH, Hamburg, Germany), a stadiometer (Seca 213, Seca GmbH, Hamburg, Germany) and a 150 mm measuring capacity slide-knot caliper (Mediworld, London, UK) respectively.

Lines 172-173 - Back up your decision with references of 2s for each phase and not 3 or more. Would this provoke a higher activation? Additionally, the word approximately was used. So did they perform each stage for 2s or not? This can be understood as some did it for 1.5s and others maybe 1.8s. Elaborate

Response: During the preliminary study which was done in our biomechanics laboratory, we found that 2s was the ideal duration for the participants to hold their heel phase positions. This also ensured that when the data was being processed it was easy to identify the phase positions from the motion data captured using the Vicon motion capture system. A 3s hold or more induced unnecessary fatigue build up across the pool of participants, hence the reason for sticking with the 2s duration with respect to each phase. All participants performed each phase hold for 2s. This was verbally counted together with the use of a metronome.  The word “approximate” has now been deleted from the sentence. Also the authors have added three references based on previous studies from our lab that have used the 2s phase hold durations.

Lines 175-176 - Reference needed to back up your decision of a 1-minute rest.

Response: Again this decision was based on our preliminary data collections within the lab. We found that a 1 minute rest was sufficient for the participants to fully recover before embarking on the next trial. Durations over 1 minute was considered too long. Three of our earlier studies have utilised similar protocols and therefore the authors have added three references based on previous studies from our lab that have also incorporated the 1 minute rest.

Line 188 - What did you do if there were 2 results close together and the 3rd had a significant difference? Did you still take the average of 3 reps or did you perform the 4th rep as this could impact your data? Elaborate

Response: The data was averaged from the collected three trials. Precautionary measured were observed by way of explaining in details the experimental protocol to our participants prior to embarking on the study. Participants were advised to produce their maximum voluntary contractions (MVC) for each of the muscles as data was collected. If a significant difference was observed in any of the trials during the MVC task or during the dynamic captured trials whilst performing the heel raise task the trial was repeated after the participant had recovered.

Results

Figure 2 - Significance in GM not marked – correct

Response: The significance in Gm is now corrected as Figure 3.

Overall your results show high SD which brings me to your sample selection and its size. Especially in PB con and ecc measurements. Elaborate

Response: With respect to the PB muscle, although precautionary measures were observed during the data collection and normalisation procedures two male participants during the concentric and eccentric muscle contractions produced high muscle activations during the heel raise dynamic task. However, overall this increment in the percentage muscle activation was not statistically significant between the female participants. The sample size is justified as the power (80%) for this study was based on earlier preliminary feasibility experiments performed in our lab using a few participants.

 Limitations of the study should be extended with the difference in the physical activity levels of participants.

Response: Thank you for your comment. I have added a sentence in the discussion to reflect your suggestion.

What is the practical application of your findings for the field? Add

Response: It is envisaged that from a non-invasive methodological approach, the EMG outputs in combination with kinetic and kinematic outcome measures could be used as a screening tool that provides valuable information for the assessment of muscles and soft tissues as they undergo rehabilitation and recovery following injury.  This statement has been added to the end of the discussion.

Kind regards,

U. Chris Ugbolue, PhD

Round 2

Reviewer 1 Report

Dear editor

The authors have made a great effort to improve the manuscript.

Best regards

Blanca

Author Response

Thank you for your comments. We are pleased we have been able to address all your comments. 

Reviewer 2 Report

Dear Authors,

Thank you for addressing the majority of my questions. However, some small parts still need some work.

2.1. Participants

Add the text that the principles of the Helsinki Declaration were followed.

The inclusion criteria are still not clear:

You wrote: ''if they had any lower limb asymmetry'' what did you mean by that? Morphological or functional? it is not clear. Elaborate and add which asymmetries did you mean.

You wrote: ''The physical activity level of all participants recruited were low to moderate'' according to which methodology (how many h/day or week)? Report and add references.

Also, does that mean that participants with high activity levels could not participate in your study? Elaborate and add info if necessary

Kind regards

Author Response

Thank you for your additional comments. Please find attached a copy of the revised manuscript with the changes indicated in red font. Below are the responses to your comments.

Comment

Thank you for addressing the majority of my questions. However, some small parts still need some work.

2.1. Participants

Add the text that the principles of the Helsinki Declaration were followed.

Response

The text has now been added.

Comment

The inclusion criteria are still not clear:

You wrote: ''if they had any lower limb asymmetry'' what did you mean by that? Morphological or functional? it is not clear. Elaborate and add which asymmetries did you mean.

Response

Thank you for your email. By lower limb asymmetry we mean one limb being shorter than the other or the uneven muscle / soft tissue distribution of one limb in comparison to the other. These factors are known to influence gait. I have now included in brackets your suggestion (i.e. morphological and functional) in the manuscript.

Comment

You wrote: ''The physical activity level of all participants recruited were low to moderate'' according to which methodology (how many h/day or week)? Report and add references.

Response

According to the American College of Sports Medicine (a complete guide to fitness and health), it was highlighted that a target of 150 min/wk in terms of duration and frequency of exercise was considered a moderate-intensity activity for maintaining good health (Bushman and American College of Sports Medicine [ACSM], 2017).

Reference: Bushman, B., and American College of Sports Medicine [ACSM] (2017). ACSM’s Complete Guide to Fitness & Health, 2E. Champaign, IL: Human Kinetics.

Comment

Also, does that mean that participants with high activity levels could not participate in your study? Elaborate and add info if necessary.

Response

Participants with higher physical activity levels of fitness and health were also considered eligible to participate in the study. This has also been added to the methods section.

Thank you once again for your constructive feedback. We hope you will approve this version of the manuscript.
